# Prevalence of Potential Pathogenic and Antimicrobial Resistant *Escherichia coli* in Danish Broilers

**DOI:** 10.3390/antibiotics12020344

**Published:** 2023-02-07

**Authors:** Louise Ladefoged Poulsen, Magne Bisgaard, Henrik Christensen

**Affiliations:** 1Department of Veterinary and Animal Sciences, University of Copenhagen, 1870 Frederiksberg, Denmark; 2Bisgaard Consulting, 4130 Viby Sjælland, Denmark

**Keywords:** broilers, broiler breeders, *E. coli*, APEC, antimicrobial resistance, virulence-associated genes

## Abstract

Avian pathogenic *Escherichia coli* (APEC) are important bacteria in broiler production in terms of economy, welfare, and use of antibiotics. During a previous outbreak of APEC in the Nordic countries, it was suggested that the pathogenic clones of *E. coli* causing the outbreak originated from grandparent stock and were transmitted to the offspring, causing increased first week mortality. This study investigated whether the pathogenic potential of *E. coli* at the parent and broiler level differs in relation to pathogenic potential described by the level of virulence-associated genes and pattern of antimicrobial resistance. The hypothesis was that, due to higher biosecurity at the parent level, the *E. coli* population will show a lower level of antimicrobial resistance and carry fewer virulence-associated genes, as a result of fewer *E. coli* infections observed. From four parent flocks and eight broiler flocks, 715 *E. coli* were isolated from cloacal swabs of newly hatched chickens (Ross 308). The isolated *E. coli* were characterized by eight virulence-associated genes and phenotypic resistance against six antimicrobials. It was found that the prevalence of virulence-associated genes and phenotypic antimicrobial resistance varied significantly between flocks, and the virulence-associated genes *papC* and *irp2* and resistance against ampicillin were significantly more prevalent in breeder flocks compared to broiler flocks.

## 1. Introduction

Avian pathogenic *Escherichia coli* (APEC) are important bacteria in broiler production, where they are associated with first week mortality (FWM), polyserositis and cellulitis in broilers [1,2] and salpingitis and peritonitis in breeders [3]. Transmission of APEC from breeders to broilers and proliferation and transmission of *E. coli* during hatch are important points in broiler production for the spread of *E. coli* [4]. The focus areas for control of FWM caused by *E. coli* and other pathogens is a high level of biosecurity and hygiene and good management of the flocks [5]. High biosecurity is also a focus area at higher levels in the production pyramid, including all processes at the hatchery and transport of broilers [5]. Despite high biosecurity and hygiene at all levels, it is not possible to eliminate pathogenic *E. coli* in broiler production, since *E. coli* is part of the commensal flora. Attempts to distinguish pathogenic *E. coli* from commensal *E. coli* are not straightforward. However, certain sequence types (ST), virulence-associated gene (VAG) profiles and serotypes may be associated with the *E. coli* most likely to cause disease [6,7,8,9]. Vertical transmission of *E. coli* from parent flocks to broilers has been shown to represent an important transmission route [10]. Therefore, it may be speculated that transmission of APEC down through the production pyramid from elite stock to great-grandparents, grandparents and broilers is of similar importance [11,12,13]. 

At the top of the production pyramid, it is anticipated that hygiene and biosecurity is higher than at the broiler level, which presumably results in fewer infections and lower use of antimicrobials among breeding animals at the top of the production pyramid [5]. This points in a direction of less use of antimicrobials at the breeder level. However, the longer lifespan of breeders may point in the opposite direction, with increased infections followed by increased use of antimicrobials. The aim of the present study was to evaluate the prevalence of antimicrobial resistant *E. coli* and potential pathogenic APEC at the grandparent and the parent stock level of Danish broiler production. The hypothesis was that, due to higher biosecurity at the grandparent level, the *E. coli* population here would show a lower level of antimicrobial resistance and hold fewer VAGs, as a result of fewer *E. coli* infections observed. Broiler breeders (grandparent offspring) and broilers (parent stock offspring) were sampled before they entered farms, to avoid environmental impact on the gut microbiota. 

*E. coli* from cloacal swabs of newly hatched chickens (Ross 308) were characterized for potential virulence-associated genes (VAGs) and phenotypic antimicrobial resistance, to evaluate whether flocks at broiler breeder level harboured *E. coli* with a lower prevalence of antimicrobial resistance and VAGs compared to birds at the broiler level. In addition, we evaluated whether specific sequence types were associated with specific broiler flocks. The identification of APEC is ambiguous and different suggestions exist for this purpose. Johnson et al. [7] suggested the identification of five plasmid-borne virulence genes (*iutA*, *hlyF*, *iss*, *iroN*, and *ompT*) in *E. coli* strains to distinguish highly pathogenic APEC from faecal *E. coli*. However, the eight genes (*papC*, *tsh*, *irp2*/*fyuA*, *iutA*/*iucD*, *iss*, *cva*/*cvi*, *astA*, and, *vat*) suggested by Ewers et al. [9] are chromosomally as well as plasmid-borne. Evidence for both protocols is strong; however, a clear definition of APEC remains to be described and may not be based on VAGs alone, but the list of VAGs may, in the future, be combined with the sequence type for an even more reliable prediction.

## 2. Results

### 2.1. Identification of Escherichia coli

In total, 720 presumptive *E. coli* isolates were included in the initial analysis. With the use of MALDI-TOF MS, it was found that 500 out of 505 tested isolates were *E. coli*. The remaining 5 isolates were identified as *Enterobacter cloacae* and originated from the same flock. These isolates were excluded from further analysis. The remaining 215 isolates were confirmed as *E. coli* by a positive indole test, thus ending up with 715 *E. coli* isolates for further analysis.

### 2.2. Prevalence of Antimicrobial Susceptibility

The prevalence of chickens in each flock carrying a resistant *E. coli* is stated in Figure 1. *E. coli* resistant to ampicillin (AMP) and sulfamethoxazole (SUL) was present in all flocks. AMP resistant *E. coli* was observed in 40% to 95% of chickens, with a mean of 70.5, while SUL resistant *E. coli* was present in 28% to 95% of chickens in the flocks (mean = 69.7). It was also common to find chickens carrying an *E. coli* resistant against tetracycline (TET), with prevalence varying from 2% (in a parent flock) to 93%, and with a lower mean value of 38.1, compared to AMP and SUL. The prevalence of chickens carrying an *E. coli* resistant against cefotaxime (CTX) and trimethoprim (TMP) varied from no chickens to 73% and 74%, with means of 21.3% and 22.8%, respectively. The prevalence of chickens carrying an *E. coli* resistant to spectinomycin (SPT) was the lowest of all antimicrobials, with no resistant *E. coli* in four chicken flocks and 28% in the flock with the highest prevalence (mean = 7.1). Significant variation between flocks was observed for all six antimicrobials (*p* values < 0.0001). 

### 2.3. Prevalence of Multidrug Resistance 

When evaluating the resistance of *E. coli* (*n* = 502) to all six antimicrobials, it was found that 1% of the isolates were resistant to none or one of the six antimicrobials, while 37% (the highest prevalence) were phenotypically resistant to four of the six antimicrobials (Figure 2A). Figure 2B shows the prevalence of *E. coli* isolates harbouring resistance against tetracycline, ampicillin, sulfamethoxazole and trimethoprim at the same time, at flock level. Tetracycline, penicillins and sulfamethoxazole, in combination with trimethoprim, were the most commonly used antimicrobials for poultry in Denmark in 2016 [14]. It was found that the prevalence of resistance to these four antimicrobials ranged from 3% in a broiler flock (196) to 51% in another broiler flock (157). The prevalence of resistance against these important antimicrobials in *E. coli* isolated from the four broiler breeder flocks included in this analysis was found to be between 31% and 35%.

In Figure 3, the percentage of *E. coli* showing phenotypical resistance to each of the six antimicrobial compounds from the four broiler breeder flocks is compared to the eight broiler flocks. The level of resistance to ampicillin was found to be significantly higher in the broiler breeders compared to the broilers (*p* = 0.0002). No significant difference was found for the remaining five antimicrobial compounds. 

### 2.4. Prevalence of Virulence-Associated Genes (VAGs)

The prevalence of chickens carrying *E. coli* harboring the VAGs *cva*/*cvi*, *vat*, *tsh*, *iucD*, *papC*, *irp2*, *iss*, or *astA* is illustrated in Figure 4A. The diagram illustrates that the prevalence of the eight VAGs varies between the flocks (*p* < 0.0001) for all genes except *astA* (*p* = 0.0008). The figure also shows that *iucD* and *iss* are the most prevalent genes of the eight genes tested for, while *papC* and *astA* are the least prevalent VAGs. The mean number of VAGs per *E. coli* isolate is illustrated in Figure 4B. A statistical difference in number of VAGs was found between flocks. For example, the two parent flocks PC and PD differed statistically from all flocks except flocks PD, PC, and CG. 

Figure 5 illustrates the prevalence of *E. coli* harbouring 0–3 and 4–8 VAGs at flock level. It is found that some flocks are colonized by *E. coli* with high numbers of VAGs. For example, in flock PC (parent flock), only 6% of the *E. coli* contain 0–3 VAGs, while 94% contain between 4–8 VAGs. In contrast, flock CB has a population of *E. coli* in which 88% of the isolates contain between 0 and 3 VAGs and 12% contain 4 to 8 VAGs. A significant flock variation was observed for all eight VAGs (*p* values < 0.0001). 

When comparing the prevalence of *E. coli* from broiler breeders versus broilers containing each of the eight VAGs (Figure 6), *papC* and *irp2* (*p* = 0.0065 and 0.0204, respectively) are found in a significantly higher prevalence in broiler breeders compared to broilers. The prevalence of the remaining six VAGs did not differ significantly between broiler breeders and broilers.

Figure 7A,B shows the prevalence of *E. coli* isolated from broiler breeders versus broilers containing either 0–3 or 4–8 VAGs. No significant difference in prevalence of VAGs could be demonstrated between breeders and broilers (0–3 VAGs: *p* = 0.1221 and 4–8 VAGs *p* = 0.1221). 

### 2.5. Multi Locus Sequence Types (MLST) and Genomic Relatedness

The phylogenetic relationship of the 70 isolates representing one parent flock (PA) and four broiler flocks (CB, CE, CG, CH) based on single nucleotide polymorphisms can be found in Figure 8. In total, 27 different MLST types (STs) were identified. ST 2040 and 69 were the most prevalent STs, making up 14% (*n* = 10) and 13% (*n* = 9) of the isolates, respectively, while ST 1946 and 117 were equally prevalent, making up 7% each (*n* = 5). ST 2040 was only found in the offspring from breeders (flock PA), while ST 69 was found in four of the five flocks, including the breeder flock. ST 1946 was found in the breeder flock and one of the broiler flocks, while ST 117 was found in two of the broiler flocks.

## 3. Discussion

During the years 2014–2016, colibacillosis increased on broiler farms in the Nordic countries [10]. A genomic comparison of *E. coli* isolates from colibacillosis from Finnish, Norwegian, and Danish broiler and broiler breeder flocks during that period concluded that, due to the high genomic similarity of the *E. coli*, a common origin of the dominant clone was most likely. Denmark imports parent stock from Swedish grandparent flocks, as do other Nordic countries. Therefore, the pathogenic *E. coli* clones circulating during 2014–2016 were concluded to originate from Swedish grandparent flocks, being vertically transmitted to parent flocks, and further down to the broiler flocks [10]. In the present study, we investigated whether the *E. coli* population colonizing broiler breeders originating from Sweden and Danish broilers may be distinguished from each other, based on in the prevalence of VAGs and antimicrobial resistance patterns. By sampling broiler breeders and broilers before exposure to the environment, the gut microbiota of newly hatched broiler breeders can be interpreted as representing the grandparents or what is transmitted from grandparents to parents, while the *E. coli* population found in broilers may represent the transmission of *E. coli* from broiler parents. Furthermore, broiler breeders and broilers are exposed to bacteria at the hatchery that may mainly be bacterial flora remaining on eggshells, and bacterial flora of other newly hatched chickens in the same hatcher, on conveyer belts and on trucks [4]. A carry-over of bacteria in hatchers from previous hatchings may occur, however cleaning and disinfection is rigorous, and this factor may be of limited importance. Due to higher biosecurity at the grandparent level compared to the parent stock level, the *E. coli* population originating from breeders may be expected to have a lower prevalence of antimicrobial resistance and presumably have a lower pathogenic potential, as indicated by a lower prevalence of VAGs. However, in this study it was found that the prevalence of antibiotic resistance and virulence genes varied significantly between flocks. A clear difference in the *E. coli* population at the breeder and the broiler level could not be identified. However, a tendency towards more virulence genes and resistance was seen in the *E. coli* population in the broiler breeder flocks.

Resistance to ampicillin was found to be significantly higher in broiler breeders compared to broilers. This finding may reflect a higher use of ampicillin in grandparent flocks, In addition, *papC* encoding P-fimbria and *irp2* encoding iron-repressible protein were significantly more prevalent in broiler breeders compared to broilers. None of the investigated antimicrobials or VAGs had a lower prevalence in the broiler breeders compared to broilers. However, a higher prevalence of *E. coli* carrying 4–8 VAGs in broiler breeders was not detected, and these have been associated with increased risk of colibacillosis [9] Therefore it cannot be suggested that *E. coli* in the broiler breeder flocks have a higher pathogenic potential as compared to the *E. coli* population in the broiler flocks in this study. The data show a high variation in prevalence of VAGs between flocks (Figure 5 and standard error of mean in Figure 6 and Figure 7), suggesting that if more flocks had been included in the study the picture may have looked different. 

The hypothesis of the broiler breeder flocks showing a lower level of antimicrobial resistance and carrying fewer VAGs could not be supported by the data in the present study, as a higher level of ampicillin resistance was found in the breeder flocks and the VAGs *papC* and *irp2* were more prevalent in the breeder flocks compared to the broiler flocks. The *papC* gene is associated with expression of the P pilus, which is important for the initial attachment of the bacteria to the host [15,16], and *irp2* is associated with iron uptake [17,18].

Broiler production faces periods with high mortality due to APEC, and it has been concluded that the breeding animals are the most likely source of the APEC causing the outbreaks [10]. Therefore, it may be suggested that, besides controlling *E. coli* by high biosecurity [5] a focus on development of methods for identification of APEC in the top layer of the production pyramid could be the initial step for development of methods to trace and combat the transmission of APEC in broiler production. Knowledge of flocks carrying APEC could then be used for reducing the risk of outbreaks, for example by lowering the stocking density.

In this study, the prevalence of resistance against AMP and SUL seemed higher than the prevalence reported by the Danish Integrated Antimicrobial Resistance Monitoring and Research Programme (DANMAP) in 2016. In the DANMAP surveillance, resistance of *E. coli* to AMP and SUL was found to be 27% for both antimicrobials [14] In this study, 30–100% of the *E. coli* was resistant to AMP and 28–87% was resistant to SUL. The difference may be explained by sampling methods. In the DANMAP report *E. coli* was isolated on a selective agar plate (Violet Red Bile Agar without antibiotic) and subsequently one *E. coli* per plate (one plate per flock) was tested for susceptibility to the antimicrobials with Sensititre^®^ plates. This method is less sensitive compared to the method used in the present study and therefore the prevalences cannot be directly compared and are expected to be higher in the present study. In Denmark, broiler flocks are not treated with antimicrobials against *E. coli*, except during outbreak situations. This is illustrated by the increased use of antimicrobials in 2015 [19] when the Nordic countries experienced high mortality due to *E. coli* in broiler production [10]. The DANMAP report shows increased use of antimicrobials again in 2020, when increased mortality was reported again (Table 4.2 [19]).

A recent study from Spain [20] reports similarly high levels of resistance against ampicillin in manure from broilers to those reported this study. Resistance against cefotaxime in *E. coli* isolated from manure was approximately 40%, compared to the present study with prevalence between 0% and 73% (mean 21.3%). Tetracycline resistance was found in 55–60% of the isolates in the Spanish study, compared to a mean of 38% in this study. Overall, there were comparable levels, however, sampling methods differed. The study from Spain used the disk diffusion method, which is less sensitive than the method used in this study. An interesting study from Italy compared the levels of antimicrobial resistance in *E. coli* from intestinal samples of broilers raised in conventional, organic and antibiotic-free production systems [21]. Resistance against ampicillin was found in 65.5%, 59.9% and 80.1% of antibiotic-free, organic and conventional farms, respectively. In this study, resistance against ampicillin was found to be 70.5% on average among all 12 flocks. Resistance against cefotaxime was found to be 4.8%, 2.6%, and 5.8% in the three different production systems used in the Italian study, compared to 21.3% in the present study. Tetracycline resistance was found in 57.6%, 55.5%, and 64.3% of the antibiotic-free, organic, and conventional farms, respectively, compared to 38% resistant *E. coli* in this study. Overall, resistance against ampicillin seemed to be comparable in the Italian study and the present study, while resistance against cefotaxime was found to be lower and resistance against tetracycline slightly higher in the Italian study. However, when comparing the different studies, it should be considered that different sampling strategies was used and different methods for testing antimicrobial resistance were also used. An Austrian study also compared the prevalence of antimicrobial resistance in organically and conventional raised broiler flocks [22]. In conventional flocks, the prevalence of resistance against sulfamethoxazole, ampicillin, trimethoprim, and tetracycline was 39.9%, 33.8%, 24.9%, and 25.9%, respectively in the conventional flocks, compared to 69.6%, 70.5%, 22.8%, and 38.1% resistance against sulfamethoxazole, ampicillin, trimethoprim, and tetracycline, respectively, in the present study. Resistance against sulfamethoxazole, ampicillin, and tetracycline is higher in Danish flocks compared to Austrian flocks, while the prevalence of trimethoprim resistance is lower in Danish flocks. In the Austrian study, the Sensititre^®^ broth microdilution system was used on single colonies, while the Spanish and Italian studies used the disk diffusion method. All three mentioned studies used methods where a single *E. coli* was isolated before antimicrobial susceptibility testing. This strategy may be anticipated to result in a lower level of antimicrobial resistance compared to the method used in this study, where 10 µL of MacConkey broth with many different *E. coli* from the cloacal swab was spread on an agar plate containing antibiotic. With this method, just a few resistant *E. coli* in the swab will result in growth on the agar plate, and resistance will be registered. Keeping the methods in mind, it may be argued that results obtained by different methods may not be comparable at all. However, when comparison is performed, it should be noted that the sensitivity of the methods differs and this is expected to influence the results.

The mean number of VAGs was also found to differ between flocks (Figure 3B). Ewers et al. [9] found that *E. coli* isolates carrying 4 to 8 of the tested VAGs were more likely to be isolated from birds with septicaemia, compared to *E. coli* carrying 0 to 3 of the VAGs. This could indicate that the flocks colonized with an *E. coli* population carrying 4–8 of the VAGs might be more prone to colibacillosis. In this study, flocks PC, PD (parent flock) and CG (broiler flock) had a high prevalence of *E. coli* harbouring four or more VAGs, indicating that these flocks may have increased risk of colibacillosis [9]. The prevalence of VAGs varied significantly between flocks and, for antimicrobial resistance, a high variation between flocks was also observed. 

The 27 different ST identified in this study have all been isolated from poultry previously, except ST 5217, which has only been uploaded to the database in relation to the present study. Several of the STs from the cloacal swabs in this study are described as APEC (ST 23, 69, 88, 93, 101, 117, 428) in the EnteroBase.

It seems that some STs are associated with specific flocks; for example, ST 2040 is found only in flock PA, ST 602 is specific to flock CG, and 5217 is specific to flock CE. Other STs seems to be more widely distributed, such as ST 69, which is present in four of the five flocks, and ST 155, which is present in three flocks. ST 69 is a common cause of urinary tract infections in humans and has also frequently been reported as pathogenic in poultry (http://enterobase.warwick.ac.uk (accessed on 9 November 2022)) [4].

Significant differences in the presence of VAGs and ST associated with colibacillosis were found between flocks, indicating that some flocks may be at increased risk of colibacillosis from the start of production. However, since no data on mortality from the flocks were included, this is a hypothesis that needs further investigation. 

## 4. Materials and Methods

### 4.1. Sampling, Study Design and Bacteriology

The study included eight broiler flocks (named CA, CB, CD, CE, CF, CG, and CH) originating from different parent stock flocks but the same Danish hatchery and four broiler breeder flocks (named PA, PB, PC, and PD) originating from the same Swedish grandparent hatchery. The breeder and broiler flocks were unrelated, but all were Ross 308 originating from Aviagen (Åstorp, Sweden). The sampling design as well as the isolation and analysis of *E. coli* is described in the flowchart in Figure 9. From broilers, 60 cloacal swabs were collected from each flock at the hatchery after quality control of the chickens. Likewise, 60 broiler breeders were sampled upon arrival at the farm, before entering the house. In total, 720 cloacal swabs were collected from the twelve flocks. Samples were collected during 2016 and 2017. Cloacal swabs were collected with thin sterile cotton swabs, which were immediately placed in a MacConkey broth (Oxoid, CM0005, Basingstoke, UK), covering the part of the cotton swab in contact with the chicken. Samples were incubated at 37 °C overnight. Subsequently, the samples were streaked on MacConkey agar plates (Oxoid, CM0007) before isolation and identification of a single *E. coli* colony per animal. Colonies were sub-cultured in Brain Heart Infusion Broth (BHI) (Oxoid, CM1135) before storage in 15% glycerol at −80 °C and used for whole genome sequencing. Isolation of *E. coli* for antimicrobial resistance typing and analysis for VAGs is described below.

### 4.2. Antimicrobial Susceptibility Testing

After pre-incubation in MacConkey broth (Oxoid, CM0005), 10 μL of each culture was streaked onto six MacConkey agar plates (Oxoid, CM0007) containing ampicillin (16 μg/mL), sulfamethoxazole (128 μg/mL), tetracycline (8 μg/mL), cefotaxime (1 μg/mL), trimethoprim (4 μg/mL), and spectinomycin (128 μg/mL), respectively. The concentration of antimicrobial compound in each plate was decided from EUCAST epidemiological cut-off values (https://www.eucast.org/, accessed on 4 November 2015) [23]. For each antimicrobial, the epidemiological cut-off value for *E. coli* was identified, and a concentration twice the cut-off value was chosen for the agar plates. Samples were incubated aerobically at 37 °C overnight. From each sample, it was registered whether growth of presumptive *E. coli* colonies (medium size, low convex, circular, and pink colonies) was present on the six different antimicrobial plates, and a single colony from each plate was isolated.

### 4.3. Identification of E. coli Strains

From each antibiotic-containing MacConkey agar plate showing growth of presumptive *E. coli* colonies, a single colony was sub-cultured on a blood agar plate (BA) with a blood agar base (Oxoid, CM0055) containing 5% calf blood and incubated at 37 °C overnight. The species identification of the first 505 isolates (505/720) was confirmed by MALDI-TOF MS (Vitek MS, Biomérieux, Marcy l´Etoile, France) as previous described [4]. For the remaining isolates, species identification was performed by colony morphology, followed by an indole test. For indole tests, isolates were grown overnight in Brain Heart Infusion Broth (BHI) (Oxoid, CM1135), to which was subsequently added 2–3 drops of Kovac’s reagent (Merch, Milipore, CAS #120-72-9, Darmstadt, Germany). A positive test appeared as an oily red layer on top of the broth. Isolates testing positive were regarded as *E. coli* and stored at −80 °C after overnight growth in Luria Broth (LB) (Oxoid, CM1023) supplemented with 15% glycerol after growth. 

### 4.4. Multidrug Susceptibility Testing

Selected *E. coli* isolates were tested for resistance against all six antimicrobials. From each flock, ten *E. coli* isolates resistant to AMP, ten isolates resistant to SUL, and so on for all six antimicrobials (AMP, SUL, TET, CTX, TMP, and SPT) were randomly selected and tested for multidrug resistance (up to 60 isolates per flock). If less than ten isolates of *E. coli* resistant against the relevant antimicrobial were obtained, all isolates were tested. From each flock, between 31 and 55 *E. coli* isolates were included, testing 501 isolates. The selected *E coli* isolates were pre-incubated in MacConkey broth at 37 °C. After incubation, 10 μL of each culture was streaked onto six MacConkey agar plates containing ampicillin (16 μg/mL), sulfamethoxazole (128 μg/mL), tetracycline (8 μg/mL), cefotaxime (1 μg/mL), trimethoprim (4 μg/mL), and spectinomycin (128 μg/mL), respectively. Samples were incubated at 37 °C overnight. From each sample it was registered whether growth was present. In the case of growth, the isolate was registered as resistant.

### 4.5. Presence of Virulence-Associated Genes

The presence of eight specific VAGs (*papC*, *iucD*, *irp2*, *tsh*, *vat*, *astA*, *iss*, and *cva*/*cvi*) was tested by a multiplex PCR [9]. The virulence-associated genes encode P-fimbria associated with adhesion [16], iron acquisition systems (*iucD* and *irp2*) [24,25], and the temperature-sensitive autotransporter protein associated with high virulence [26]. The *vat* gene encodes a cytotoxic vacuolating autotransporter protein [27] the enteroaggregative heat-stable enterotoxin (*astA*) [28] a protein for increased serum survival (*iss*) [29] and the plasmid-borne genes *cva*/*cvi* which cause disruption of the membrane of sensitive cells [30]. *E. coli* isolated from MacConkey without any antimicrobials was used (Oxoid, CM0007). In brief, DNA was extracted by using a Maxwell^®^ RSC Instrument and Cultured Cells DNA kits (AS1620, Maxwell, Nacka, Sweden), as recommended by the manufacturer. PCR running conditions [9] were followed, except that agarose gels were run for 80 min at 75 V. A GeneRuler 100 bp plus DNA ladder (Thermo Scientific, SM0323, Vilnius, Lithuania) was used as a size marker on each gel.

### 4.6. Statistics

For comparison of the prevalence of newly hatched chickens carrying *E. coli* resistant to AMP, SUL, TET, CTX, TMP, and SPT, and *E. coli* carrying the VAGs; *papC*, *iucD*, *irp2*, *tsh*, *vat*, *astA*, *iss*, and *cva*/*cvi*, GraphPad Prism, version 6 (Graph Pad software Inc., San Diego, CA, USA) was used for Chi-square tests and Fisher’s exact test for larger and smaller sample sets, respectively. Differences were considered significant if *p* was <0.05. For comparison of difference in resistance and number of VAGs between broiler breeder and broilers, values were calculated as fractions of analysed samples and evaluated by t-tests with standard error of means (SEM) (GraphPad Prism, version 6 (Graph Pad software Inc.)).

### 4.7. Genome Analysis

From one broiler breeder flock and four broiler flocks, 70 *E. coli* isolates obtained from MacConkey agar without antimicrobials were selected for whole genome sequencing. These *E. coli* were isolated from the same cloacal swab as the isolates as used for phenotypic antimicrobial susceptibility testing and VAG analysis, but may potentially be different *E. coli*, since they were isolated on MacConkey agar without supplementation of antimicrobials. From the breeder flock, 23 isolates were randomly selected, while 10–13 isolates were randomly selected from each of the broiler flocks. DNA was extracted using a Maxwell^®^ FSC robot with the Maxwell^®^ RSC Cultured Cells DNA Kit (Promega, AS1620, Nacka, Sweden). The libraries were prepared from separate aliquots of gDNA samples using an Illumina Nextera DNA kit with modifications. In brief, the gDNA was first quantified using a Qubit dsDNA HS Assay Kit (Thermo Fisher Scientific, Scoresby, Australia). All gDNA concentrations were standardized to the same concentration to achieve uniform reaction efficiency in the tagmentation step. Standard Illumina Nextera adaptors were used for sample tagmentation. PCR-mediated adapter addition and library amplification were carried out using customized indexed i5 and i7 adaptor primers (IDT, Coralville, IA, USA), which were developed based on the standard Nextera XT indexed i5 and i7 adapters (e.g., N701-N729 and S502-S522). Libraries were then pooled and size selected using SPRI-Select magnetic beads (Beckman Coulter, Lane Cove West, Australia). Finally, the pooled library was quality checked and quantified on an Agilent Bioanalyzer 2100, using the DNA HS kit (Agilent, Santa Clara, CA, USA). Whole genome sequencing was performed using an Illumina HiSeq 2500 v4 sequencer in rapid PE150 mode (Illumina, San Diego, CA, USA). Sequence read quality was initially assessed using FastQC version 0.11.5 (http://www.bioinformatics.babraham.ac.uk/projects/fastqc/, accessed 24 January 2018). Illumina raw reads passing quality control were assembled into draft genome sequences using the A5 assembly pipeline version A5-miseq 20150522.

The multi locus sequence types of the isolates were determined by uploading assembled reads to the MLST1.8 server https://cge.cbs.dtu.dk/services/MLST/ (accessed on 13 September 2018) [31].

In order to investigate the relatedness of *E. coli* with identical sequence types (ST) of different flock origin, a single nucleotide polymorphism analysis (SNP) was performed using CSI Phylogeny 1.4 provided by the Danish Technical University at https://cge.cbs.dtu.dk/services/CSIPhylogeny/ (accessed on 8 October 2018), where *E. coli* K12, MG1655 (U00096.3) was used as the reference genome. Software from iTOL was used for visualization of genetic relatedness [32].

## 5. Conclusions

The prevalence of VAGs and antimicrobial resistance in *E. coli* isolated from cloacal swabs from parent and broiler flocks varied between flocks. In addition, a significantly higher prevalence of *E. coli* resistant to ampicillin in broiler breeder flocks and a higher prevalence of *E. coli* holding *papC* and *irp2* genes in these flocks were demontrated.

Despite the difference in resistance to ampicillin and difference in the prevalence of two of the eight VAGs between breeder and broiler flocks, it may not be concluded that the prevalence of presumptive APEC differs between the flock types. To draw this conclusion a higher number of flocks should be included in the investigation, as well as production data including first week mortality and total mortality in the production period. Mortality rates, combined with post mortem examinations of dead breeders/broilers and bacteriological culturing to determine whether the cause of death relates to *E. coli*, are necessary to decide whether APEC populations differ between the different types of flocks. However, it can be concluded that the prevalence of presumptive avian pathogenic *E. coli* varies considerably in both breeder and broiler flocks.

## Figures and Tables

**Figure 1 antibiotics-12-00344-f001:**
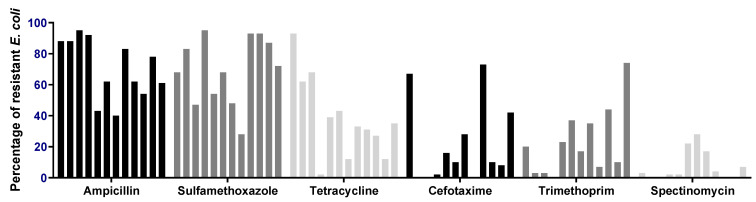
The percentage of chickens carrying an *E. coli* resistant to six different antimicrobials. Each bar represents one flock. The first four bars in each category/antimicrobial represent the 4 broiler breeder flocks and the remaining bars represent the 8 broiler flocks.

**Figure 2 antibiotics-12-00344-f002:**
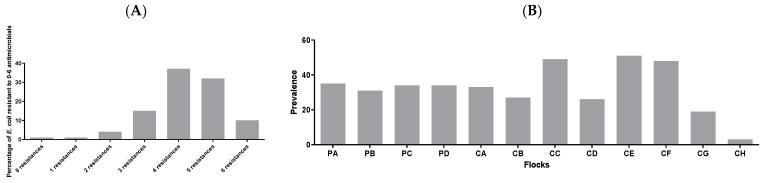
(**A**) Number of phenotypical resistances of 502 *E. coli* isolated from 12 flocks; (**B**) Prevalence of *E. coli* (*n* = 502) that are resistant against tetracycline, ampicillin, sulfamethoxazole, and trimethoprim.

**Figure 3 antibiotics-12-00344-f003:**
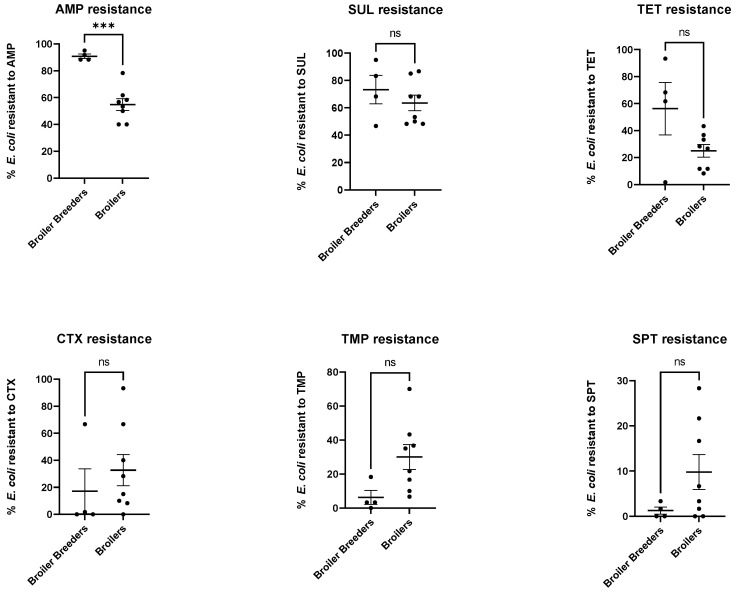
Scatter dot plots showing mean with SEM comparing the fraction of isolates showing phenotypic resistance towards each of the antimicrobial compounds in the four broiler breeder flocks compared to the eight broiler flocks. ***: *p* < 0.001, ns = non-significant.

**Figure 4 antibiotics-12-00344-f004:**
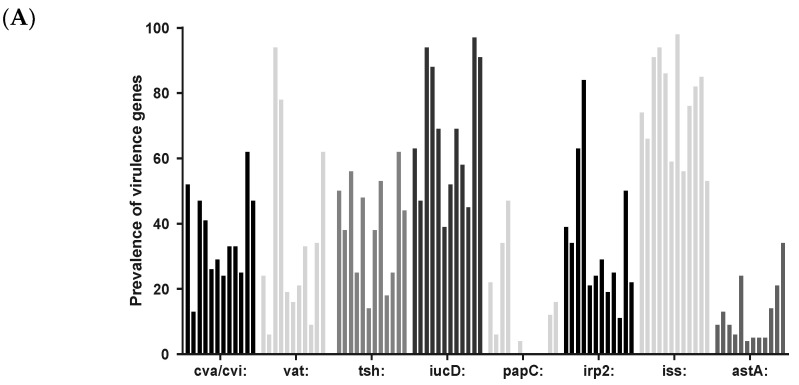
(**A**) The prevalence of chickens that carry an *E. coli* with one of the eight different virulence-associated genes (*cva*/*cvi*, *vat*, *tsh*, *iucD*, *papC*, *irp2*, *iss*, and *astA*). Each bar represents one flock. The first four bars are the broiler breeder flocks and the remaining eight bars are the broiler flocks. (**B**) Mean number of virulence-associated genes per *E. coli* per flock (the first four bars represent the breeder flocks).

**Figure 5 antibiotics-12-00344-f005:**
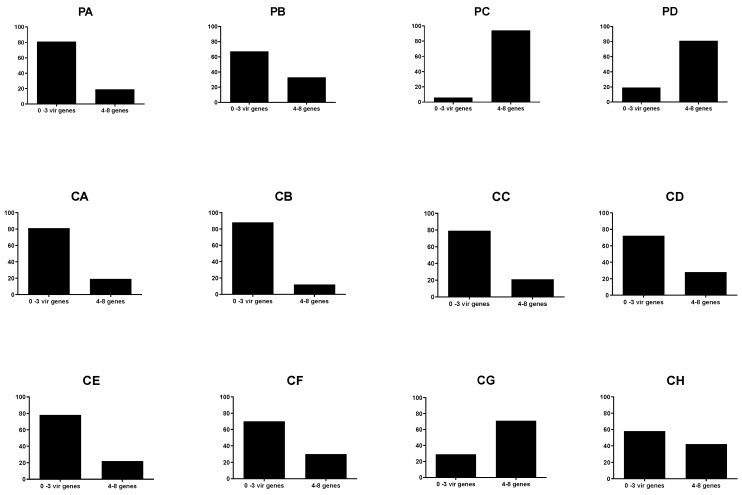
Prevalence of *E. coli* harboring 0–3 and 4–8 virulence-associated genes at flock level. The flocks PA, PB, PC, and PD are parent flocks, and the remaining 8 flocks are broiler flocks.

**Figure 6 antibiotics-12-00344-f006:**
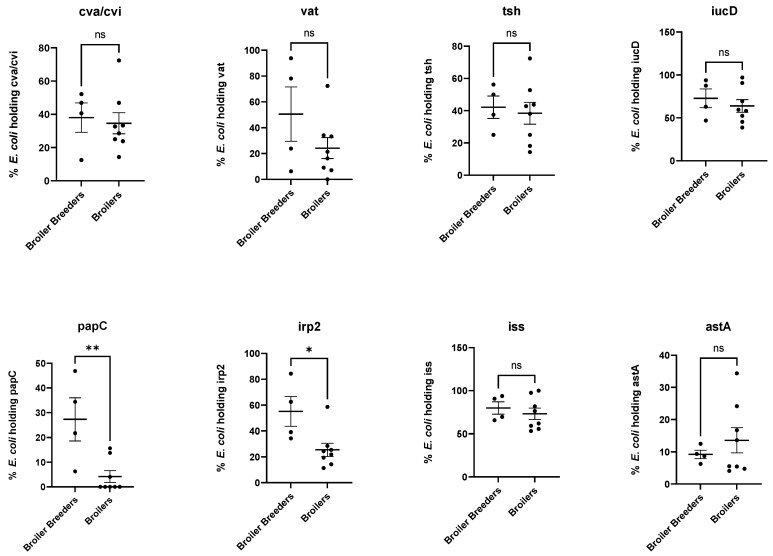
Scatter dot plots showing mean with SEM, comparing the fraction of isolates harbouring each of the eight virulence genes in the four broiler breeder flocks compared to the eight broiler flocks. *: *p* < 0.05, **: *p* < 0.01, ns = non-significant.

**Figure 7 antibiotics-12-00344-f007:**
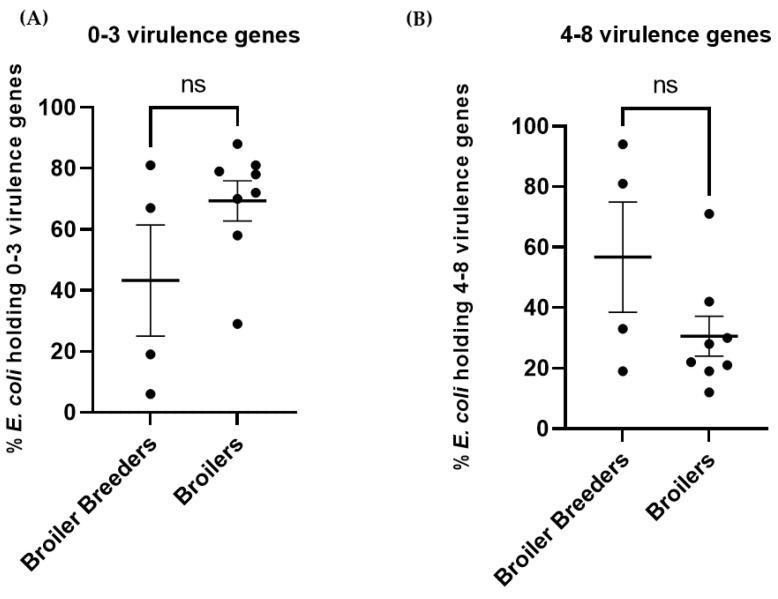
(**A**). Scatter dot plot demonstrating the mean (with SEM) percentage of *E. coli* containing 0–3 VAGs isolated from broiler breeders versus broilers. (**B**) Scatter dot plot demonstrating the percentage of *E. coli* containing 4–8 VAGs isolated from broiler breeders versus broilers. ns = non-significant.

**Figure 8 antibiotics-12-00344-f008:**
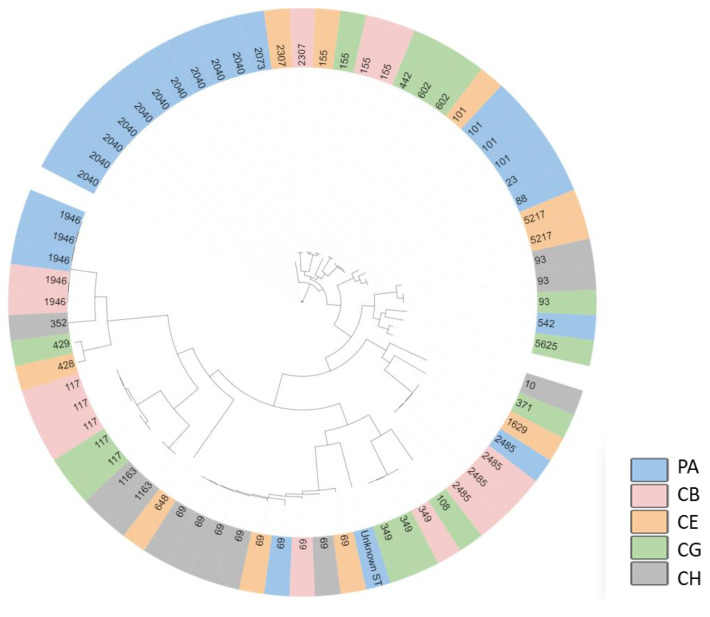
Phylogenetic relationship of 70 *E. coli* based on single nucleotide polymorphism. Each isolate is named by the MLST type and colors refer to flock number. The *E. coli* originate from one breeder flock (flock PA) and four broiler flocks.

**Figure 9 antibiotics-12-00344-f009:**
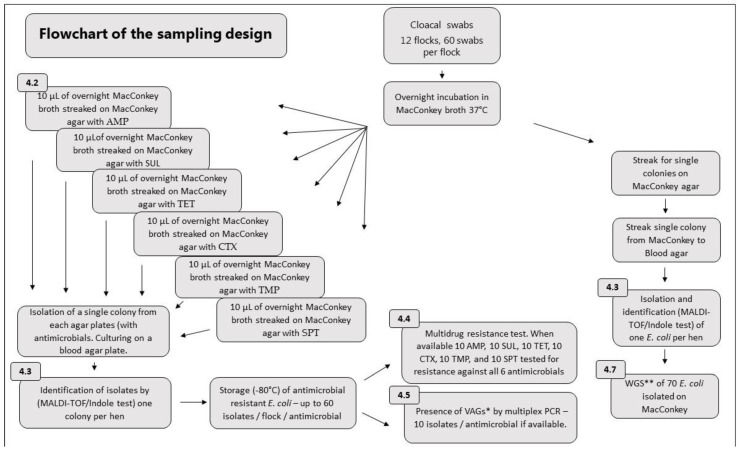
Flowchart describing the sampling design, isolation and analysis of *E. coli.* AMP = Ampicillin, SUL = Sulfamethoxazole, TET = Tetracycline, CTX = Cefotaxime, TMP = Trimethoprim, SPT = spectinomycin. * VAG = Virulence-associated genes, ** WGS = Whole genome sequencing.

## Data Availability

Not applicable.

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
