# Peer review of "Prevalence of Potential Pathogenic and Antimicrobial Resistant Escherichia coli in Danish Broilers"

_antibiotics, 2023, doi:10.3390/antibiotics12020344_

Round 1

Reviewer 1 Report

Abstract

1) line 9: please put the acronym (APEC).

Introduction

2) lines 52-55:

3) line 57: since the study was based on cloacal swabs, “gut microbiota” is more appropriate than “microflora”.

4) lines 52-55 and 61-65: these sentences are redundant. Please, remove one or add the work conclusion to summarize the article.

Materials and Methods

5) line 263: The authors do not present information regarding ethical authorizations in animal experimentation. I recommend not publishing the article without a statement from the authors or the collection authorization code from the animal ethics committee, which seems more appropriate.

6) line 296: I have doubts about the remaining isolates and the specificity of the culture-dependent method to infer that they are E. coli since the MacConkey media and the indole test was not exclusively for E.coli. Wouldn't it be more appropriate to incubate the isolates at temperatures higher than 40°C since E. coli is thermotolerant to increase the chances of certainty?

7) line 310: at what temperature?

8) line 317, please include the function of each gene screened in this study.

Discussion

9) lines 178-183: please re-write the sentence and put the summary conclusions found in this work.

Conclusion

10) lines 372-376: please re-write the sentence more concisely.

General

11) Please improve the figure’s resolution and put the standard deviation.

12) Please put all “E. coli” in italics.

Author Response

I have uploaded the response to reviewer as a pdf

Reviewer 2 Report

Line 60: Please add the brief description about virulence genes to improve the scope and interest of the reader in the manuscript. The criterial behind inclusion of only 8 genes for characterization purpose need to be elaborated. Also, add little detail related to commonly used virulence gene to characterize pathogenic E. coli of avian origin e.g. as described by Johnson et al. 2008; Awawdeh et al. 2018; Grakh et al. 2022 in different regions around the world.  

Line 67: Although detail is mentioned in the material and method section but as the results section precedes, the authors may try to add the detail of the sample numbers as they are different in next subsections. e.g. line 68 to 70, there is change from 505 to 215. This might hamper the readability and interest of the reader.

Line 67-70: Please specify the reasons behind the confirmation of few isolates by Indole test.  

Line 73: The abbreviated forms used in the text for antibiotics are as per the standards of manufacturer or agencies, however, the abbreviations need to be expanded in the first appearance in the text within manuscript. E.g. expand the AMP, SUL and other according to their first appearance in the text.

Line 125: Close the parentheses

Line 370-372: Despite the difference in resistance to ampicillin and difference in the prevalence of two of the eight VAGs between breeder and broiler flocks, it’s difficult to conclude that the prevalence of presumptive APEC differs between the flock types. The breeder flock might not be the reason for the transfer for the APEC/MDR E. coli to broiler flocks. If so, list or expand some other factors that might be responsible for such resistance among broiler flocks. The other possible factors like management or lack of biosecurity/hygiene/litter build-up etc need to be discussed. The related part may be added to discussion and may also be included in the conclusion section.

Author Response

We have uploaded our response to reviewer 2 as a pdf attachment

Reviewer 3 Report

Dear authors,

I think the topic of your work is quite interesting. As a matter of fact, colibacillosis is probably the most important avian disease in terms of economic losses, and the interest in Avian Pathogenic Escherichia Coli is increasing. I especially found appealing that the present study focuses on the vertical transmission of APEC and ultimately on first week mortality matters. Moreover, the geographical framework helps the reader to understand something about poultry production in Northern Europe.

Unfortunately, I think major revisions are needed before the work can be published, as I have some alarming concerns about the design of the study and the Materials and Methods part.

I will start my report by listing some general minor issues along with abstract and introduction matters.

Then, I will share my main concern; at last I will list few points about results and discussion, as I couldn't fully take those part into account.

Firstly, I am not a big fan of the use of first person (singular and plural) in scientific articles. Though, as I don’t think it’s an actual error, it is your choice to change it or not through the manuscript.

I think in some cases you directly used an acronym without writing the full definition first (e.g.: AMP and SUL, I cannot retrieve the first association of acronym and full definition).

ABSTRACT:

Line 18: a comma between "flock" and "715", or rephrase;

Line 18-20: sounds like all E.coli had 8 virulence genes, maybe rephrase to improve sentence clarity.

INTRODUCTION:

lines 34-35: The control is not only for E.coli but for FWM in general.

lines 37-40: It should be rephrased as the sentence sound a little too complex.

lines 40-44: It should be rephrased as the sentence sound a little too complex.

Lines 55-65: maybe these facts should be transposed to the discussion part, especially the lines 55-59.

MATERIALS AND METHODS

Lines 284-285: add EUCAST in references.

Line 317: here I would talk about testing for the presence of VAGs rather than the prevalence.

I think the materials and methods are not clear in general, or may have some serious flaws. As a matter of facts, it sounds like the E.coli isolated on MacConkey plates and tested in parts 4.5 and 4.7 (VAGs and WGS) are different from those tested in part 4.2, 4.3 and 4.4. As the microflora of a day-old chick can be composed of more than one kind of E.coli, it is not predictable that you tested the same E.coli. You might have tested different E.coli from the same animal: I think it is not clear why you need phenotypic resistance from a strain and genotypic resistance from another. As you didn’t mention this possibility anywhere else, I think some explanations are needed in order to further revise this manuscript. Maybe the revision of the whole manuscript, especially the Materials and Methods part, may help to improve the clarity of study design and true aims. I attached a flowchart about how I interpreted the materials and methods.

DISCUSSION

I think that references can be improved here, by searching some more studies (at least 4 or 5) from Europe as a comparison. For example:

lines 226-238: this part is based solely on a review: maybe it should be changed with data from more Northern European countries, or at least other countries with high level of poultry industry such as Poland, France or Italy. I would ignore studies from other continents, as they bring along too many differences in antibiotic use, poultry management, etc. There is no need for that.

Lastly, you should consider and mention the fact that there can be horizontal transmission at hatchery, thus an infection with E.coli from different groups can happen. You never acknowledged this possibility.

Author Response

It seems that I am not able to upload two documents here. So I have merged the response to reviewer with a flowchart of the methods, which I would like to include in the manuscript as a supplementary.

Please note that the line numbers corresponds to the linenumbers found when the manuscript are in track-change mode with All markup.

Round 2

Reviewer 1 Report

The authors answered all questions, and the suggestions were implemented in this new manuscript version. The authors also justified the animal experimentation ethics committee, as shown in the response letter. Thus I recommend this version of the manuscript for publication.

Author Response

The answer is attached as pdf

Reviewer 3 Report

Dear authors,

thank you for considering all my suggestions and comments. Let’s start from the major concern, i.e. Materials and Methods. I think the single paragraph were indeed clear, but it was very difficult to point out the common thread of the procedures. Therefore, I strongly agree with a flowchart in the supplementary files. The one you are suggesting is perfect, and will help young scientist to promptly understand the research methods.

Moreover, I think that it should be stated somewhere in the discussion section that the E.coli isolated and tested for antimicrobial resistance are potentially different from those isolated and tested for VAGs and WGS. This notion is partly displayed in lines 299-306 (“This strategy…influence the results.), but another clearer statement would be appreciated. I am insisting on this topic because for example the discussion paragraph “The hypothesis of the broiler…iron uptake.” (lines 225-231) makes a comparison of results coming from potentially different E.coli. In general, I think that acknowledging this notion would improve the manuscript’s scientific relevance.

As I already started with discussion, I think that moving part of the introduction here was an improvement for the manuscript. Moreover:

Lines 197-202: Thank you for adding this notion: please support the sentences with reference/references.

lines 207-209: sentence should be simplified: "...not be identified although..." changed with "...not be identified: though,..." or "...not be identified, although..." or whatever you like the most.

Line 251: “[21]” was not cancelled, hope it doesn't mess up the whole references.

Lines 308-316: “the mean number of…was also observed.” maybe it should be moved somewhere else, before the antimicrobial susceptibility.

Line 330: change “include” to “included”.

Introduction:

Lines 68-76: sentences “The identification of APEC…prediction.” are fine in the intro, but maybe they should be moved elsewhere before the aim.

Results:

line 105: non à none

lines 109-111: sentence "Tetracycline...in 2016" fits better in discussion.

lines 111: “It is found that…” change the verb tense to past.

Author Response

The answers has been attached as a pdf file
